# Comparison of the Analgesic Effect of Ropivacaine with Fentanyl and Ropivacaine Alone in Continuous Epidural Infusion for Acute Herpes Zoster Management: A Retrospective Study

**DOI:** 10.3390/medicina56010022

**Published:** 2020-01-08

**Authors:** Hee Yong Kang, Chung Hun Lee, Sang Sik Choi, Mi Kyoung Lee, Yeon Joo Lee, Jong Sun Park

**Affiliations:** 1Department of Anesthesiology and Pain Medicine, Kyung Hee University Hospital, Kyungheedae Road 23, Dongdaemun-Gu, Seoul 02447, Korea; ujuabba@gmail.com; 2Department of Anesthesiology and Pain Medicine, Korea University Medical Center, Guro Hospital, Gurodong Road 148, Guro-Gu, Seoul 08308, Korea; clonidine@empal.com (S.S.C.); mknim@hotmail.com (M.K.L.); pipipipig@naver.com (Y.J.L.); pkjgsn@naver.com (J.S.P.)

**Keywords:** continuous epidural infusion, fentanyl, herpes zoster, ropivacaine, postherpetic neuralgia, neuropathic pain, intervention

## Abstract

*Background and Objectives*: Currently, few studies have reported the effects of opioids during continuous epidural infusion (CEI) to control pain owing to herpes zoster (HZ). This study aimed to retrospectively compare the effectiveness of epidural opioids in the treatment of acute HZ pain. *Materials and Methods*: We reviewed medical records of 105 patients who were divided into two groups: R group (CEI with ropivacaine) and RF group (CEI with ropivacaine and fentanyl). Clinical efficacy was evaluated using the numeric rating scale (NRS) score for 6 months after the procedures. We compared the percentage of patients with complete remission in each group. We investigated the complication rates during CEI. *Results*: No significant differences in the NRS scores were observed between the two groups in the 6-month period. The adjusted odds ratio (OR) for patients included in the complete remission was 0.6 times lower in the RF group than in the R group (95% confidence interval: 0.22–1.71, *p* = 0.35). The OR for complications during CEI was higher in the RF group than in the R group. However, the difference was not statistically significant. *Conclusions*: No difference was observed in the management of HZ pain and the prevention of postherpetic neuralgia between the two groups. The incidence of complications tended to be higher in the RF group than in the R group.

## 1. Introduction

Neuropathic pain may be due to multiple pathologic conditions such as herpes zoster (HZ), diabetes, cancer, and radiculopathy and affects the quality of life of elderly patients [1]. In particular, pain owing to HZ, caused by the reactivation of the varicella-zoster virus (VZV), is known to be more prevalent in elderly patients with reduced T-cell immunity [2]. Pain associated with HZ is defined as acute herpetic neuralgia until 30 days after the skin rash, subacute herpetic neuralgia until 3 months after the acute phase, and postherpetic neuralgia (PHN) afterward [3]. PHN can persist for more than 12 months, and PHN can last for 6 months in 8.5% of patients and for 12 months in 6% of patients, despite early treatment with antiviral drugs [4,5]. Continuous or intermittent zoster-associated pain can also affect the quality of life and the ability to work. Therefore, effective treatments for zoster-associated pain (ZAP) are necessary to prevent negative sequelae. 

Medications such as antiepileptics, tricyclic antidepressants, lidocaine patches, and analgesics may be effective in relieving pain [6,7]. However, interventional therapy is recommended for uncontrolled pain or in the case of a high likelihood of transition to PHN, even with sufficient medication [6,7]. Epidural analgesia, paravertebral block, and pulsed radiofrequency have been performed as interventional therapies to reduce pain [6,7,8]. Recently, it has been reported that continuous epidural infusion (CEI) using a catheter is effective for ZAP [9,10,11]. Most studies have shown that epidural administration of local anesthetics is effective in controlling ZAP [9,10,11]. However, few studies have compared the effectiveness of epidural opioids (particularly fentanyl) in controlling pain owing to HZ.

Thus, this retrospective study aimed to determine whether the addition of opioids to local anesthetics during CEI in patients with acute HZ showed synergistic effects with previously known local anesthetics and opioids.

## 2. Materials and Methods

### 2.1. Study Design

This study was performed after the approval by the Institutional Review Board of Korea University Medical Center, Guro Hospital, Seoul, Republic of Korea (2018GR0057, 8 February 2018). The study also complied with the STROBE checklist (Appendix A). The medical records of patients who underwent CEI for acute HZ from April 2014 to July 2018 were collected. Patients were divided into two groups: R group, in which patients were administered CEI containing ropivacaine without fentanyl, and RF group, in which patients were administered CEI containing ropivacaine with fentanyl. Inclusion criteria were patients older than 50 years with a numeric rating scale (NRS; 0 = no pain and 10 = worst possible pain) score of ≥4; patients who received only the standard drug therapy, including antiviral agents, until the administration of CEI; and patients who underwent follow-up for a period of 6 months after CEI. Exclusion criteria were patients with insufficient medical records, patients who did not maintain the catheter for more than 10 days after CEI, patients who did not receive standard medications such as antiviral agents until the procedure, and patients who discontinued the prescribed standard medication (gabapentin or pregabalin, and analgesics such as oxycodone) owing to the side effects of the drugs. Patients who underwent other procedures for pain control within 6 months after the procedure were excluded from this study but were included in the analysis of complications during the catheter maintenance period. Complication rates during the catheter maintenance period were separately analyzed.

### 2.2. Procedure 

After placing the patient in the prone position, an aseptic dressing was applied to the procedure site. A Tuohy epidural needle (18-gauge) was inserted into the interlaminar space at the third or fourth level below the target level under fluoroscopic guidance. The loss of resistance technique was used to verify if the needle was located in the epidural space. After confirming the needle position, a 20-gauge epidural catheter (EpiStim™, length: 800 mm, Sewoon Medical Co., Ltd, Seoul, Korea) was placed at the target level through the Tuohy needle (Figure 1) with radiographic confirmation. This type of epidural catheter has a built-in conductive guidewire (Nitinol, length = 1100 mm) with 800 mm inside the catheter and 300 mm exposed for connection to an electric nerve stimulator. The cathode of the electric nerve stimulator (Life-Tech EZstim, Stafford, TX, USA) was connected to the exposed guidewire, and the anode was attached to an electrode on the patient’s calf. A zero to 5 mA electric current was then delivered through the guidewire. Verbal communication with the patient confirmed that the electric stimulation had reached the HZ-affected area. The catheter was placed in the appropriate epidural space, and once electric stimulation was initiated, verbal communication with the patient confirmed the sensation. After patient confirmation, further communication established that the electric stimulus was following the HZ dermatome. If the electric stimulation was in a region other than the HZ dermatome, the epidural catheter was adjusted under fluoroscopy, and the electric stimulation was repeated to confirm its presence in the HZ-affected area. Once the stimulus was established in the correct area, the guidewire was removed. After removing the guidewire, the position of the epidural catheter tip was confirmed using a contrast agent and fluoroscopy; 0.19% ropivacaine and 1 mg dexamethasone (total 8 mL) were then administered via an epidural catheter in the R group, and 0.19% ropivacaine and 1 mg dexamethasone with 50 μg fentanyl (total 8 mL) were administered in the RF group. After initial drug injection, patients in the R group were administered CEI (total 275 mL) containing 30–45 mL of 0.75% ropivacaine (37.1 ± 4.1) and normal saline without fentanyl. Patients in the RF group were administered CEI (total 275 mL) containing 30–40 mL of 0.75% ropivacaine (36.1 ± 3.7) and normal saline with 200 μg fentanyl. In both groups, CEI was administered at a rate of 4 mL/h via a portable balloon infusion device (AutoFuser pump, ACE Medical Co., Ltd., Seoul, Korea). Concentrations of ropivacaine in the portable balloon infusion device were adjusted according to the degree of pain relief or side effects. In both groups, the epidural catheter was fixed by subcutaneous tunneling to decrease the risk of infection and catheter migration. The inserted catheter was maintained in its position for a minimum of 10 days and was removed after 2 weeks. Whether patients were undergoing catheterization as inpatients or outpatients, a physician changed the dressings daily and monitored the procedure site. Additionally, antiepileptics (pregabalin or gabapentin) and analgesics were administered to patients in both groups. Antiepileptics were prescribed by adjusting the drug dose according to age and renal function and were tapered according to symptoms. Oral oxycodone was administered as an analgesic, starting with the minimum reported effective dose for PHN [12].

### 2.3. Data Collection

Data regarding age, sex, spinal level of catheterization, days from the onset of rash to CEI, past medical history (hypertension (HTN), diabetes mellitus (DM), liver disease, kidney disease, and asthma), and amount of 0.75% ropivacaine in the infusion device were collected. The patients’ pain was assessed using an 11-point verbal NRS (0, no pain and 10, unbearable pain). We collected the NRS score data at different times: immediately before the procedure (baseline NRS score); 1 h after the procedure; 14 days after the procedure; and 1, 3, and 6 months after the procedure from patients’ medical records. Complete remission was defined as an NRS score ≤2, with no further medication prescribed at 6 months after the procedure. The number of patients included in this category during the 6-month follow-up period was recorded. We recorded whether other interventional procedures were performed owing to inadequate pain control during the 6-month follow-up period after CEI. We also collected records of complications (nausea, vomiting, dysuria, hypotension, and itching) during the epidural catheter maintenance period.

### 2.4. Outcome Measures

To compare analgesic effect as the primary endpoint, we compared NRS scores of the two groups at baseline; 1 h; 14 days; and 1, 3, and 6 months after the procedure, after correcting for various variables. As the secondary endpoint, we compared the percentage of patients with complete remission in each group, the proportion of patients requiring additional nerve block owing to inadequate pain control after the procedure, and the incidence of complications owing to CEI between the two groups.

### 2.5. Statistical Analysis

Demographic data were analyzed using the Kolmogorov–Smirnov test to assess the normality of the distribution. Normally distributed sets of demographic data were compared between the groups using an independent *t*-test, and non-normally distributed datasets were compared using the Mann–Whitney U test. After correcting for various confounding variables (age, sex, spinal level of catheterization, days from the onset of rash to CEI, past medical history (HTN, DM, liver disease, kidney disease, and asthma), average amount of 0.75% ropivacaine in the infusion device, and baseline NRS score), we analyzed the differences in NRS scores between the groups using analysis of covariance. Multivariable logistic regression analysis was used to compare the percentages of complete remission within 6 months after the procedure and to compare the percentages of patients who underwent other interventional treatments between the two groups. Univariable logistic regression analysis was performed to compare the complication rates owing to CEI. Data are presented as mean ± standard deviation or median (interquartile range). The collected data were analyzed using Statistical Package for the Social Sciences (SPSS) software (version 17.0; SPSS 157 Inc., Chicago, IL, USA). All statistical tests were two-sided, and the threshold for statistical significance was set at *p* < 0.05.

## 3. Results

We reviewed the medical records of 137 patients. Five patients missed the follow-up or had inadequate medical records in the 6 months after the procedure. Twelve patients underwent other interventional procedures within 6 months of CEI. In six patients, the catheter could not be maintained for more than 10 days owing to side effects associated with CEI. Two patients did not receive antiviral drugs at the beginning of the HZ episode. During the 6-month follow-up period, one patient had other pain-causing diseases. The medical records of these patients were excluded from the final analysis. Moreover, six patients discontinued using antiepileptics or analgesics owing to drug-associated side effects after the procedure. To prevent drug-induced bias, these patients were also excluded from the final analysis. After the exclusions, the medical records of the remaining 105 patients were analyzed. Sixty-two patients were assigned to the R group and 43 were assigned to the RF group (Figure 2).

No significant differences were observed in the demographic and clinical characteristics of the patients between the two groups (Table 1, Appendix A).

No significant difference was observed between the two groups with respect to post-procedure NRS scores after correcting for confounding variables (Table 2). There was no significant difference in the reduction ratio of NRS scores according to fentanyl use with time after CEI (Figure 3). 

The proportion of patients showing complete remission was lower in the RF group than in the R group. However, the difference was not statistically significant (adjusted odds ratio (OR): 0.62, 95% confidence interval (CI): 0.22–1.71, *p* = 0.35; Table 3). The proportion of patients who underwent other interventional procedures owing to insufficient pain control within 6 months after CEI was 3.21 times higher in the RF group than in the R group (adjusted OR: 3.21, 95% CI: 0.56–18.24, *p* = 0.19; Table 4).

Complication rates (nausea, vomiting, dysuria, hypotension, and itching) during CEI were higher in the RF group than in the R group (Table 5). However, these differences were not statistically significant.

## 4. Discussion

This study determined whether there was a difference in pain reduction and complication rates between local anesthetics alone and local anesthetics with opioids in CEI for patients with acute HZ. No significant difference was observed in pain reduction between the two groups during the 6-month follow-up period. Furthermore, there was no significant difference in the complete remission rate and the proportion of patients who underwent other invasive procedures owing to inadequate pain control within 6 months after CEI. The rates of complications such as nausea (unadjusted OR: 6.98), vomiting (unadjusted OR: 4.02), itching (unadjusted OR: 9.50), and hypotension (unadjusted OR: 9.50) during CEI were consistently higher in the RF group than in the R group. However, the difference was not statistically significant.

Reactivated VZV in the sensory ganglia of the spinal cord manifests as HZ and subsequently spreads out to the affected dermatome, thereby producing an inflammatory response and inducing nerve damage. A serious initial nerve damage or inability to restore normal function after the loss of nerve function can lead to PHN [13]. Therefore, active treatment before the nerve damage occurs can help prevent PHN and control pain. A recent meta-analysis reported that CEI in the acute phase of HZ is effective in controlling pain and preventing PHN [14]. The rationale behind using CEI to manage acute HZ pain and prevent PHN is that the interruption of afferent pain stimuli to the central nervous system and the improved blood flow to nerve tissue will minimize neural damage and reduce pain to the patient by blocking sympathetic nerves [15,16].

The analgesic effect of epidural opioids added to local anesthetics has been proven in several studies [17,18,19,20,21]. Neuraxial local anesthetics and opioids function synergistically to provide analgesia during labor [17,18]. For postoperative pain control, the addition of fentanyl to 0.1% solution of ropivacaine enhances analgesic efficacy [19]. The coadministration of fentanyl (100 μg) and 1% ropivacaine accelerated the onset of sensory and motor blocks during epidural ropivacaine [20] or lidocaine [21] anesthesia, without significant side effects related to fentanyl. The analgesic effect of epidural opioids seems to be related to the location of receptor sites in the spinal cord. These receptor sites have been identified in many areas of the brain and central nervous system, with very high densities in the substantia gelatinosa of the spinal cord [22]. Previous studies on epidural labor analgesia [17,18] and postoperative epidural analgesia [19] have also reported that opioid administration in combination with local anesthesia in the epidural space has a better effect on pain control than the administration of local anesthetics alone. However, the pain control measures in these studies were primarily performed for nociceptive pain.

Some studies have suggested that the effects of opioid-mediated analgesia in neuropathic pain may be reduced by an increase in neuropeptide cholecystokinin, an endogenous inhibitor of opioid-mediated analgesia [23]. Moreover, in neuropathic pain, the functional pool of opioid receptors is likely to be reduced at the spinal level [24].

These factors suggest that the addition of opioids during CEI for ZAP, which is a type of neuropathic pain, may not be more effective than the local anesthetic alone. Watt et al. concluded that after epidural morphine administration in 11 PHN patients, the morphine was more likely to cause side effects than pain relief [25]. In our study, no difference was observed in pain reduction between epidural infusion using local anesthetic alone and epidural infusion using local anesthetic and fentanyl to control ZAP.

Epidural opioid administration can cause several systemic side effects [26,27]. In the present study, the incidence of side effects such as nausea, vomiting, dysuria, and itching tended to be higher in the RF group than in the R group. Previous studies have reported no difference in the rate of complications with epidural opioid administration [17,18,19,20,21]. However, in these studies, only young participants were included in the trial. The average age of HZ patients included in the present study was 65.8 years. In other words, the higher incidence of complications in the opioid group in the present study may be because of the older age of patients.

Previous studies have reported that a single epidural block may be effective in managing ZAP; however, it has limited efficacy in preventing PHN [28,29]. Hence, patients who could not maintain the CEI catheter for more than 10 days were excluded from the analysis in the present study.

All patients included in the present study underwent CEI and received simultaneous antiepileptic and analgesic medications. To avoid bias owing to drug treatments, patients who discontinued the drugs owing to side effects were excluded from the analysis. 

The complete remission rate in the present study was 74% in the R group and 70% in the RF group. Reportedly, the greater severity of acute ZAP is associated with a greater likelihood of its progression to PHN [30,31]. In our hospital, interventional treatments such as CEI were performed only for patients with HZ who had an NRS score ≥4 even after oral medication. All patients included in this study had NRS scores ≥4 (mean 7.2 ± 1.3, 7.3 ± 1.0). The higher NRS scores could be one of the reasons for the lower rates of complete remission. Additionally, the definition we adopted for complete remission (NRS score ≤2, no further medication prescribed) could possibly be another reason for the lower remission rates, as other studies have defined complete remission as a pain-free state with an NRS score <3 or have not mentioned the withdrawal of medication [11,28]. 

Although CEI shows excellent pain control effect in HZ patients, it is invasive; thus, there is a possibility of side effects (epidural hematoma, epidural abscess, etc.). However, none of the patients in this study reported epidural infection, possibly because of daily dressing by well-trained physicians and well-educated patients and caregivers. A previous study [6] reported a low risk of epidural hematoma associated with epidural blocks, and in this study, epidural hematoma was not reported in patients. However, it should be implemented considering risks and benefits.

This study had some limitations in the interpretation of the results. Since this was a retrospective study, unmeasured confounding variables may have influenced the results. However, to control potential confounding variables, we performed an analysis of covariance with the baseline demographics and underlying diseases of patients as covariates. We excluded patients who were treated with other interventional procedures within 6 months after CEI. This approach may have caused a selection bias in this study. However, inclusion of these patients might have resulted in uncertainty regarding whether the improvement in the patients’ symptoms was because of CEI or other interventions. We excluded patients who received other interventions when calculating the complete remission rates and 6-month pain scores and separately analyzed the ratios. To overcome the selection bias, we selected only patients who underwent CEI with the same type of catheter (EpiStim™ epidural catheter) and those for whom CEI was performed by the same anesthesiologist. The study showed no significant difference between the two groups in terms of the degree of pain reduction or the complication rates owing to the limited number of patients. However, the incidence of complications was consistently higher in the RF group than in the R group. A larger sample size may result in statistically significant results, especially with respect to the complication rates. In the present study, patients were not randomly assigned to the groups because we analyzed medical records during normal practice. The patients’ medical records were written without blinding. Therefore, the present study may have an inherent risk of bias.

## 5. Conclusions

No difference was observed in the management of ZAP and the prevention of PHN between the group with added opioid and the one without opioid in the local anesthetic during CEI. The difference in the incidence of complications was not significant; however, it was consistently higher in the group with opioid added to CEI. A well-planned prospective study with a greater number of patients to compare strategies for preventing ZAP and PHN is needed to validate the results of this study.

## Figures and Tables

**Figure 1 medicina-56-00022-f001:**
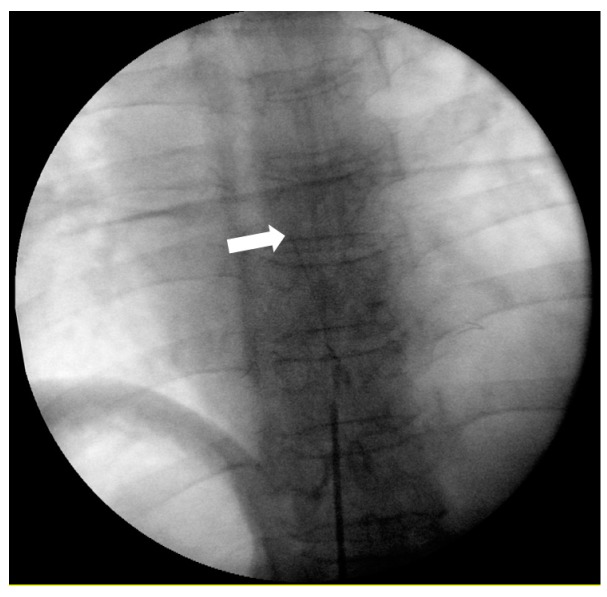
Fluoroscopic images of the continuous epidural block. The catheter used for the procedure had a built-in conductive guidewire that allowed the detection of the catheter tip using radiography, along with electrical stimulation. The arrow indicates the tip of the guidewire in the EpiStim™ catheter.

**Figure 2 medicina-56-00022-f002:**
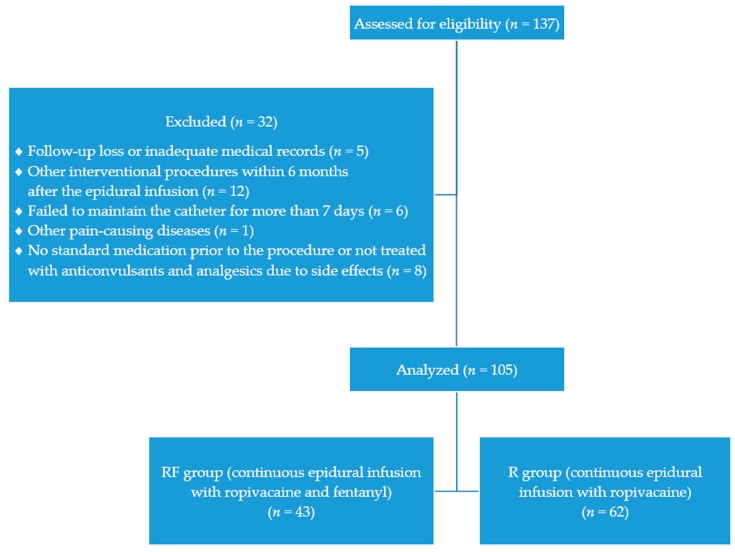
Flow diagram showing patient inclusion.

**Figure 3 medicina-56-00022-f003:**
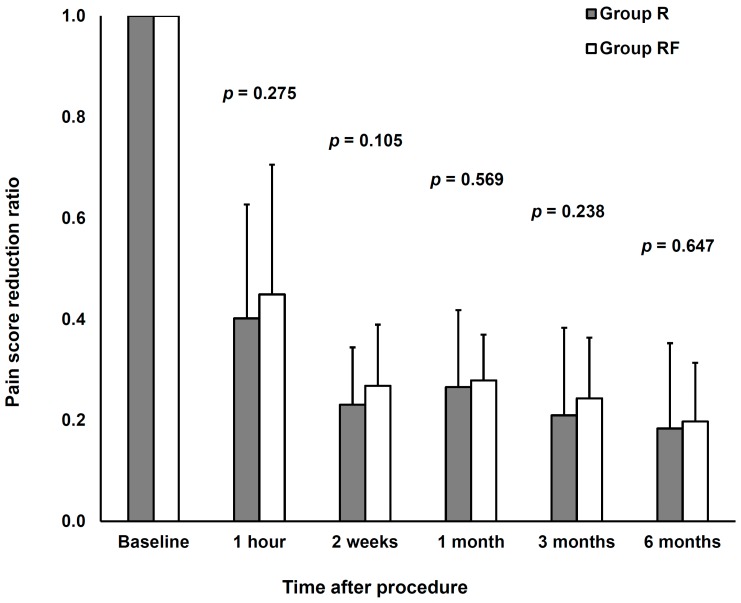
Comparison of the numeric rating scale score reduction ratio according to fentanyl use with time after continuous epidural infusion.

**Table 1 medicina-56-00022-t001:** Baseline demographic and clinical characteristics of the patients.

Demographic and Clinical Characteristics	Acute HZ (≤30 Days) R Group (*n* = 62)	Acute HZ (≤30 Days) RF Group (*n* = 43)	*p* Value
Age (years)	66.2 ± 9.3	65.3 ± 7.9	0.61
Sex (male/female)	20/42	13/30	0.84
Site of HZ infection	C: 14 (22.6%)	C: 7 (16.3%)	0.87
	T: 36 (58.1%)	T: 25 (58.1%)	
	L: 12 (19.4%)	L: 11 (25.6%)	
HTN	23 (37%)	20 (47%)	0.42
DM	18 (29%)	14 (33%)	0.83
Asthma	2 (3%)	1 (2%)	1.0
Hepatic disease	3 (5%)	1 (2%)	0.64
Kidney disease	3 (5%)	2 (5%)	1.0
Baseline NRS score	8 (6–8)	7 (6.5–8)	0.87
Avg. amount of 0.75% ropivacaine in infusion device (mL)	37.1 ± 4.1	36.1 ± 3.7	0.25

HZ: herpes zoster, Avg: average, HTN: hypertension, DM: diabetes mellitus, NRS: numeric rating scale of 0–10, C: cervical, T: thoracic, L: lumbar, R group: local anesthetic without epidural opioid, RF group: local anesthetic with epidural opioid. Data are presented as mean ± standard deviation, median (interquartile range), or number (%).

**Table 2 medicina-56-00022-t002:** Comparison of numeric rating scale scores between the groups after correction for confounding variables.

NRS Score	Acute HZ (≤30 Days) R Group (*n* = 62)	Acute HZ (≤30 Days) RF Group (*n* = 43)	*p* Value
Baseline NRS score	7.1 ± 1.5	7.3 ± 1.0	0.31
NRS score * 1 h after epidural procedure	2.9 ± 1.8	3.1 ± 1.3	0.26
NRS score * 14 days after epidural procedure	1.7 ± 0.8	1.9 ± 0.8	0.16
NRS score * 1 month after epidural procedure	1.9 ± 1.1	2.1 ± 0.7	0.18
NRS score * 3 months after epidural procedure	1.5 ± 1.1	1.8 ± 0.8	0.11
NRS score * 6 months after epidural procedure	1.2 ± 1.0	1.4 ± 0.9	0.18

* The NRS scores (0 = no pain and 10 = worst possible pain) are represented as means ± standard deviation, R group: local anesthetic without epidural opioid, RF group: local anesthetic with epidural opioid. Data were analyzed for the difference in the pain scores between the groups using analysis of covariance. Adjustments were made for age, sex, time from the onset of rash to epidural infusion, average amount of 0.75% ropivacaine in the infusion device, location of herpes zoster, hypertension, diabetes mellitus, asthma, hepatic disease, and kidney disease.

**Table 3 medicina-56-00022-t003:** Comparison of complete remission during the 6-month follow-up period between the groups after epidural infusion.

Complete Remission	R Group	RF Group	Adjusted OR (95% CI) Reference: R Group	*p* Value
Acute HZ (≤30 days)	46/62 (74%)	30/43 (70%)	0.62 (0.22–1.71)	0.35

OR: odds ratio, CI: confidence interval, R group: local anesthetic without epidural opioid, RF group: local anesthetic with epidural opioid. Complete remission is defined as a numerical rating scale score of <2 with no further medication. Data are presented as number (%). Data were analyzed using multivariable logistic regression analysis. Adjustments were made for age, sex, location of herpes zoster, days from the onset of rash to epidural infusion, average amount of 0.75% ropivacaine in the infusion device, history of hypertension, diabetes mellitus, asthma, hepatic disease, and kidney disease, and baseline pain score.

**Table 4 medicina-56-00022-t004:** Comparison of other interventions performed owing to insufficient pain control during the 6-month follow-up period after epidural infusion.

Performed Other Interventions	R Group	RF Group	Adjusted OR (95% CI) Reference: R Group	*p* Value
Acute HZ (≤30 days)	6/68 (9%)	6/49 (12%)	3.21 (0.56–18.24)	0.19

R group: local anesthetic without epidural opioid, RF group: local anesthetic with epidural opioid. Data are presented as number (%). Data included patients who underwent other procedures within the 6-month follow-up period and those who did not maintain the epidural catheter for more than 7 days owing to the side effects. Data were analyzed using multivariable logistic regression analysis. Adjustments were made for age, sex, location of herpes zoster, days from the onset of rash to epidural infusion, average amount of 0.75% ropivacaine in infusion device, history of hypertension, diabetes mellitus, asthma, hepatic disease, and kidney disease, and baseline pain score.

**Table 5 medicina-56-00022-t005:** Comparison of complication rates during continuous epidural infusion between the groups.

Side Effects	R Group	RF Group	Unadjusted OR (95% CI) Reference: R Group	*p* Value
Nausea	1/68 (1%)	5/53 (9%)	6.98 (0.79–61.67)	0.08
Vomiting	1/68 (1%)	3/53 (6%)	4.02 (0.41–39.80)	0.23
Dysuria	2/68 (3%)	3/53 (6%)	1.98 (0.32–12.30)	0.46
Itching sensation	0/68 (0%)	3/53 (6%)	9.50 (0.48–187.95)	0.14
Hypotension	0/68 (0%)	3/53 (6%)	9.50 (0.48–187.95)	0.14

R group: local anesthetic without epidural opioid, RF group: local anesthetic with epidural opioid. Data are presented as number (%). Data included patients who underwent other procedures within the 6-month follow-up period and those who did not maintain the epidural catheter for more than 7 days owing to the side effects. Data were analyzed using univariable logistic regression analysis.

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
