# Peer review of "Comparison of the Analgesic Effect of Ropivacaine with Fentanyl and Ropivacaine Alone in Continuous Epidural Infusion for Acute Herpes Zoster Management: A Retrospective Study"

_medicina, 2020, doi:10.3390/medicina56010022_

Round 1

Reviewer 1 Report

Yes, treating severe HZ pain and preventing PHN is totally desirable. The concept of the manuscript is of high interested and it is generally well written. A few comments and suggestions below:

- Abstract:

SHOULD more clearly state what what the aim of the study was: effectiveness in the treatment of acute pain (HZ) or over 6 months? 

- Introduction:

Line 37: it was just stated that PHN means pain persistence for at least 3 months and now it said that "PHN can persist for 1-12 months and zoster-associated pain (ZAP) can last for 6 months...". Being consistent in using the terms is necessary. What is more than 3 months if PHN. In addition, PHN can persistent for more than 12 months (years)

- Experimental Section

2.1Study design: line 61- please define what numeric rating scale is

Line 66-67: It is not mentioned that opioids analgesics intake by the patients was also collected (if this is the case)

2.2 Procedure

It is not clear what ropivacaine concentration was used (0.11-0.15% or 0.75% as mentioned in the rest of the manuscript). Please clarify

Line 105-106: where the patients taking oral oxycodone (or other oral opioids) compared with those not using oral opioids?

2.4 Outcome measure

Please define what the main outcome measure was: the complete remission at 6 months or the comparison of NRS at various time points

- Results 

Table 2: state that the numeric rating score are represented as "means" +/- standard deviation 

-Discussion

May discuss that the results of this study suggest that addition of opioids in the CEI is not necessary or indicated

Should probably discuss that oral opioids possibly taken by some patients may represent a confounding factor (not addressed in the methods or results)

May also discuss (even briefly) the potential risks and complications of CEI itself (vs conservative care) 

Author Response

MEDICINA-649356

Comparison of the analgesic effect of ropivacaine with fentanyl and ropivacaine alone in continuous epidural infusion for acute herpes zoster management: a retrospective study

Medicina

Response to reviewer

We sincerely appreciate the reviewer’s comments and suggestions. We have revised the manuscript according to the recommendations and believe that the comments and suggestions have greatly improved the quality of our manuscript. Our point-by-point responses and details of the corresponding changes in the manuscript are listed below.

Sincerely,

Chung Hun Lee

Department of Anesthesiology and Pain medicine,

Korea University Medical Center, Guro Hospital,

Gurodong Road 148, Guro-Gu, Seoul 08308, Republic of Korea

E-mail: bodlch@naver.com

Review Comments to the Author

Reviewer: Yes, treating severe HZ pain and preventing PHN is totally desirable. The concept of the manuscript is of high interested and it is generally well written. A few comments and suggestions below:

- Abstract:

SHOULD more clearly state what the aim of the study was: effectiveness in the treatment of acute pain (HZ) or over 6 months?

-> Thank you for your comment. As suggested, for further clarity, we have revised the purpose of the research in the Abstract section.

Lines 15-16: “This study aimed to retrospectively compare the effectiveness of epidural opioids in the treatment of acute HZ pain.”

- Introduction:

Line 37: it was just stated that PHN means pain persistence for at least 3 months and now it said that "PHN can persist for 1-12 months and zoster-associated pain (ZAP) can last for 6 months...". Being consistent in using the terms is necessary. What is more than 3 months if PHN. In addition, PHN can persistent for more than 12 months (years)

-> Thank you for your comment. As suggested, for clarity, the PHN period has been modified in the Introduction section, and the terminology has been consistently changed.

Lines 37-39: “PHN can persist for more than 12 months, and PHN can last for 6 months in 8.5% of patients and for 12 months in 6% of patients, despite early treatment with antiviral drugs.”

- Experimental Section

2.1Study design: line 61- please define what numeric rating scale is

-> Thank you for your comment. As the reviewer suggested, the numeric rating scale is described in detail as a measure of pain.

Lines 63-63: “Inclusion criteria were patients older than 50 years with a numeric rating scale (NRS; 0 = no pain and 10 = worst possible pain) score of ≥4.”

Line 66-67: It is not mentioned that opioids analgesics intake by the patients was also collected (if this is the case)

-> Thank you for your comment. As suggested, we have added the phrase “analgesics such as oxycodone” on line 68.

2.2 Procedure

It is not clear what ropivacaine concentration was used (0.11-0.15% or 0.75% as mentioned in the rest of the manuscript). Please clarify

-> Thank you for your comment. The ropivacaine we used was actually prepared at a concentration of 0.75% (7.5 mg/ml). The portable balloon infusion device was filled with 30-45 ml of 0.75% ropivacaine, and the rest was filled with normal saline to obtain a total volume of 275 ml. This mixture was then continuously administered to the patient via the epidural catheter at a rate of 4 ml/h. However, as you mentioned, readers may find it confusing to refer to diluted concentrations of ropivacaine in normal saline and the amount of 0.75% ropivacaine in a portable balloon infusion device. Therefore, we revised the text to describe the uniform amount of 0.75% ropivacaine in the portable balloon infusion device.

Lines 95-98: “After initial drug injection, patients in the R group were administered CEI (total 275 ml) containing 30–45 ml of 0.75% ropivacaine (37.1 ± 4.1) and normal saline without fentanyl. Patients in the RF group were administered CEI (total 275 ml) containing 30–40 ml of 0.75% ropivacaine (36.1 ± 3.7) and normal saline with 200 μg fentanyl.”

Line 105-106: where the patients taking oral oxycodone (or other oral opioids) compared with those not using oral opioids?

-> Thank you for your comment. We prescribed a minimum dose of oral oxycodone to all patients who underwent CEI owing to acute herpes zoster pain. To avoid bias owing to oral drug treatments, patients who discontinued oral oxycodone owing to side effects were excluded from the analysis. We have added the term “oral” on line 108 to reduce confusion.

2.4 Outcome measure

Please define what the main outcome measure was: the complete remission at 6 months or the comparison of NRS at various time points

-> As suggested, we have included the definition of main outcome on lines 129-134.

Lines 129-134: “To compare analgesic effect as the primary endpoint, we compared NRS scores of the two groups at baseline; 1 hour; 14 days; and 1, 3, and 6 months after the procedure, after correcting for various variables. As the secondary endpoint, we compared the percentage of patients with complete remission in each group, the proportion of patients requiring additional nerve block owing to inadequate pain control after the procedure, and the incidence of complications owing to CEI between the two groups.”

- Results

Table 2: state that the numeric rating score are represented as "means" +/- standard deviation

-> Thank you for your comment. In Table 2, the sentence has been modified based on your suggestion.

“NRS: numeric rating scale, *: The NRS scores (0 = no pain and 10 = worst possible pain) are represented as means ± standard deviation”

-Discussion

May discuss that the results of this study suggest that addition of opioids in the CEI is not necessary or indicated

Should probably discuss that oral opioids possibly taken by some patients may represent a confounding factor (not addressed in the methods or results)

May also discuss (even briefly) the potential risks and complications of CEI itself (vs conservative care)

-> As per your comment, the study results suggest that additional epidural administration of opioids in ZAP-related CEI is not required or indicated. Oral opioids such as oxycodone administered orlay may be confounding in the study, and patients who discontinued oral oxycodone owing to side effects were excluded from the analysis. A brief discussion regarding the potential risks and complications of CEI compared with those of conservative treatment is included in the discussion section.

Lines 285-290: “Although CEI shows excellent pain control effect in HZ patients, it is invasive; thus, there is a possibility of side effects (epidural hematoma, epidural abscess, etc.). However, none of the patients in this study reported epidural infection, possibly because of daily dressing by well-trained physicians and well-educated patients and caregivers. A previous study [6] reported a low risk of epidural hematoma associated with epidural blocks, and in this study, epidural hematoma was not reported in patients. However, it should be implemented considering risks and benefits.

Reviewer 2 Report

Kang et al. performed a study to compare the analgesic effect of ropivacaine with fentanyl versus ropivacaine alone in continuous epidural infusion for acute herpes zoster management. No herpes zoster pain relieve between two treatment was found but higher complications in the ropivacaine with fentanyl treatment than ropivacaine alone. Some concerns was noted.

Major comments

This is a small sample size study, so no significant difference with wide range confidence interval observed between the two groups could be expected. The author also mentioned that a larger sample size may result in statistically significant results, especially with respect to the complication rates. Did the author calculate the estimated sample size to evaluate the difference between two treatment?

The conclusion in the abstract was that the incidence of complications was consistently higher in the RF group. However, the rates of complications such as nausea (unadjusted OR: 6.98), vomiting (unadjusted OR: 4.02), itching (unadjusted OR: 9.50), and hypotension (unadjusted OR: 226 9.50) during CEI were was not statistically significant. In addition, unadjusted OR was presented here, so how to make sure the higher complications rate in the RF group than R group?

As mention in the introduction, interventional therapy is recommended for uncontrolled pain or in the case of a high likelihood of transition to PHN, so what is the rationale for these patients with herpes zoster under continuous epidural infusion treatment initially?

Minor comments

In Table 1, Table 3, Table 4, and Table 5, the author can present the number (%) without 95% confidence interval.

Author Response

MEDICINA-649356

Comparison of the analgesic effect of ropivacaine with fentanyl and ropivacaine alone in continuous epidural infusion for acute herpes zoster management: a retrospective study

Medicina

Response to reviewer

We sincerely appreciate the reviewer’s comments and suggestions. We have revised the manuscript according to the recommendations and believe that the comments and suggestions have greatly improved the quality of our manuscript. Our point-by-point responses and details of the corresponding changes in the manuscript are listed below.

Sincerely,

Chung Hun Lee

Department of Anesthesiology and Pain medicine,

Korea University Medical Center, Guro Hospital,

Gurodong Road 148, Guro-Gu, Seoul 08308, Republic of Korea

E-mail: bodlch@naver.com

Review Comments to the Author

Reviewer: Kang et al. performed a study to compare the analgesic effect of ropivacaine with fentanyl versus ropivacaine alone in continuous epidural infusion for acute herpes zoster management. No herpes zoster pain relieve between two treatment was found but higher complications in the ropivacaine with fentanyl treatment than ropivacaine alone. Some concerns was noted.

Major comments

This is a small sample size study, so no significant difference with wide range confidence interval observed between the two groups could be expected. The author also mentioned that a larger sample size may result in statistically significant results, especially with respect to the complication rates. Did the author calculate the estimated sample size to evaluate the difference between two treatment?

-> Because this study was retrospective in design, we did not calculate an estimated sample size to assess the difference between the two treatments. However, patient records from 2014 to 2018 were collected through maximal total surveys of patients with acute herpes zoster who underwent CEI and who did not control pain with medication. In this study, the complication rate was not statistically significant, although it was consistently higher in the RF group than in the R group. Based on these results, we estimated that a larger sample size could yield statistically significant results with regard to complication rates. To supplement this, a prospective research based on our study will be needed.

The conclusion in the abstract was that the incidence of complications was consistently higher in the RF group. However, the rates of complications such as nausea (unadjusted OR: 6.98), vomiting (unadjusted OR: 4.02), itching (unadjusted OR: 9.50), and hypotension (unadjusted OR: 9.50) during CEI were was not statistically significant. In addition, unadjusted OR was presented here, so how to make sure the higher complications rate in the RF group than R group?

-> Thank you for your comment. As you indicated, the complication incidence between the two groups was not statistically significant. In addition, the unadjusted OR was presented in the text because the number of patients with complications was small (no patients had itching sensation or hypotension in the R group). However, the complication rates that we investigated were not significant but consistently high in the RF group. Therefore, in the conclusion of the abstract, we stated that the incidence of complications was consistently higher in the RF group. However, as you stated, there was no significant difference in the incidence of complications between the two groups; hence, the conclusion of the abstract was revised to “The incidence of complications tended to be higher in the RF group than in the R group.”

As mention in the introduction, interventional therapy is recommended for uncontrolled pain or in the case of a high likelihood of transition to PHN, so what is the rationale for these patients with herpes zoster under continuous epidural infusion treatment initially?

-> Previous studies have reported that CEI reduces pain in acute shingles and reduces the transition to PHN.[1, 2] Furthermore, compared with oral medication alone, CEI more effectively reduces pain and the transition to PHN.[3]Reactivated VZV in the sensory ganglia of the spinal cord manifests as HZ and subsequently spreads out to the affected dermatome, thereby producing an inflammatory response and inducing nerve damage. A serious initial nerve damage or inability to restore normal function after the loss of nerve function can lead to PHN.[6] Therefore, active treatment before nerve damage occurs can help prevent PHN and can control pain.

The rationale behind using CEI to manage acute HZ pain and prevent PHN is that the interruption of afferent pain stimuli to the central nervous system and the improved blood flow to nerve tissue will minimize neural damage and will reduce pain in patients by blocking the sympathetic nerves.[4, 5] Epidural administration of local anesthetics interferes with the sensitization process by blocking the sympathetic nerves with their analgesic action. This action can reduce the occurrence of nerve damage and the resulting neuropathic pain.[5]

Minor comments

In Table 1, Table 3, Table 4, and Table 5, the author can present the number (%) without 95% confidence interval.

-> Thank you for your comment. As you mentioned, Tables 1, 3, 4, and 5 only provided number (%) with no 95% confidence interval.

References

Pasqualucci, A.; Pasqualucci, V.; Galla, F.; De Angelis, V.; Marzocchi, V.; Colussi, R.; Paoletti, F.; Girardis, M.; Lugano, M.; Del Sindaco, F. Prevention of post-herpetic neuralgia: acyclovir and prednisolone versus epidural local anesthetic and methylprednisolone. Acta Anaesthesiol. Scand. 2000, 44, 910-918. Kim, H.J.; Ahn, H.S.; Lee, J.Y.; Choi, S.S.; Cheong, Y.S.; Kwon, K.; Yoon, S.H.; Leem, J.G. Effects of applying nerve blocks to prevent postherpetic neuralgia in patients with acute herpes zoster: a systematic review and meta-analysis. Korean J. Pain. 2017, 30, 3. Seo, Y.G,; Kim, S.H.; Choi, S.S.; Lee, M.K.; Lee, C.H.; Kim, J.E. Effectiveness of continuous epidural analgesia on acute herpes zoster and postherpetic neuralgia: a retrospective study. Medicine (Baltimore). 2018, 97, e9837. Doran, C.; Yi, X. The anti-inflammatory effect of local anesthetics. Pain Clin. 2007, 19, 207-213. Opstelten, W.; van Wijck, A.J.; Stolker, R.J. Interventions to prevent postherpetic neuralgia: cutaneous and percutaneous techniques. Pain. 2004, 107, 202-206. Winnie, A.P.; Hartwell, P.W. Relationship between time of treatment of acute herpes zoster with sympathetic blockade and prevention of postherpetic neuralgia: clinical support for a new theory of the mechanism by which sympathetic blockade provides therapeutic benefit. Reg. Anesth. 1993, 18, 277-282.

Round 2

Reviewer 2 Report

All the comments had been addressed. I have no more question for this manuscript.